# Effects of Avocado Oil Supplementation on Insulin Sensitivity, Cognition, and Inflammatory and Oxidative Stress Markers in Different Tissues of Diet-Induced Obese Mice

**DOI:** 10.3390/nu14142906

**Published:** 2022-07-15

**Authors:** Schérolin de Oliveira Marques, Alexandre Pastoris Muller, Thais Fernandes Luciano, Natália dos Santos Tramontin, Mateus da Silva Caetano, Bruno Luis da Silva Pieri, Tatiane Lima Amorim, Marcone Augusto Leal de Oliveira, Cláudio Teodoro de Souza

**Affiliations:** 1Laboratory of Biomedicine Translational, Extreme South of Santa Catarina University, Criciúma 88806-000, SC, Brazil; scherimarques@hotmail.com (S.d.O.M.); alexandrep.muller@gmail.com (A.P.M.); bionutrithais@gmail.com (T.F.L.); nataliastramontin@hotmail.com (N.d.S.T.); mateus.caetano98@hotmail.com (M.d.S.C.); pierinutri@gmail.com (B.L.d.S.P.); 2Department of Pharmacology, Santa Catarina Federal University, Florianópolis 88040-900, SC, Brazil; 3Analytical Chemistry and Chemometrics Group, Department of Chemistry, Institute of Exact Sciences, Juiz de Fora Federal University, Juiz de Fora 36036-900, MG, Brazil; tatianelamorim@gmail.com (T.L.A.); marcone.oliveira@ufjf.edu.br (M.A.L.d.O.); 4Department of Internal Medicine, Medicine School, Juiz de Fora Federal University, Juiz de Fora 36038-330, MG, Brazil

**Keywords:** obesity, reactive oxygen species, cytokines, memory, insulin resistance, avocado oil supplementation

## Abstract

Obesity induces insulin resistance, chronic inflammation, oxidative stress, and neurocognitive impairment. Avocado oil (AO) has antioxidants and anti-inflammatory effects. This study evaluated the effect of AO supplementation on obese mice in the adipose tissue, muscle, liver, and hippocampus. Male C57BL/6J mice received a standard and high-fat diet (20 weeks) and then were supplemented with AO (4 mL/kg of body weight, 90 days) and divided into the following groups: control (control), control + avocado oil (control + AO), diet-induced obesity (DIO), and diet-induced obesity + avocado oil (DIO + AO) (*n* = 10/group). AO supplementation was found to improve insulin sensitivity and decrease hepatic fat accumulation and serum triglyceride levels in DIO mice. AO improved cognitive performance and did not affect mood parameters. Oxidative marker levels were decreased in DIO + AO mice in all the tissues and were concomitant with increased catalase and superoxide dismutase activities in the epididymal adipose tissue and quadriceps, as well as increased catalase activity in the liver. AO in obese animals further induced reductions in TNF-α and IL-1β expressions in the epididymal adipose tissue and quadriceps. These results suggest that AO supplementation has the potential to be an effective strategy for combating the effects of obesity in rats, and human studies are needed to confirm these findings.

## 1. Introduction

Obesity is increasing in prevalence and is associated with various morbid conditions [1]. One of the negative impacts of increased body fat occurs in the sensitivity to insulin action [2]. Two of the possible mechanisms underlying insulin resistance (IR) are the production of reactive species [3] and the development of associated chronic subclinical inflammation [4].

Adipose tissue may undergo rapid hypertrophy due to excessive accumulation of triglycerides in response to food overload, leading to its remodeling, followed by macrophage infiltration with increased inflammation and pro-inflammatory adipokine production, such as tumor necrosis factor-α (TNF-α) and interleukin 1β (IL-1β), accompanied by increased free fatty acid release [4,5], resulting in subclinical chronic inflammation [6]. Moreover, hypertrophied adipocytes and hyperglycemia are significant sources of ROS and reactive nitrogen species (RNS) [1]. Obese rats present a high degree of lipid peroxidation and hydrogen peroxide production in adipose tissue, while they exhibit reduced levels and activity of antioxidant enzymes, such as superoxide dismutase (SOD), glutathione peroxidase (GPX), and catalase (CAT) [7]. Moreover, these peripheral metabolic changes induce neuroinflammation and oxidative stress in the brain, associated with impaired cognitive performance and mood alterations [8]. Diets rich in saturated fatty acids induce ROS generation, cytokine production, and increase biomarkers of cell damage, and can therefore prevent or treat obesity and its associated conditions [1,9,10].

Modulating oxidative stress and inflammation can help find new ways to identify, prevent, and treat obesity and its complications. As an alternative for the treatment of obesity and its associated conditions, a variety of plants and fruits, which exert positive effects on lipid and glucose metabolism, such as the avocado, are studied [11]. Avocado oil has high levels of antioxidants, including polyphenols, proanthocyanins, and tocopherols, showing beneficial health results. Studies in humans and rodents show that avocado helps weight loss, reduces the risk of diabetes [12], normalizes blood cholesterol levels [13], is involved in hepatic metabolism [14], and exhibits anti-inflammatory properties [15]. In diabetic rats, avocado oil (AO) supplementation decreases ROS production and brain damage [16]. Nonetheless, the protective effects of avocado oil on hepatic lipids, insulin resistance, damage to the brain, and inflammatory and oxidative stress profiles in obese people are still unknown. Thus, the present study aims to examine whether avocado oil supplementation could suppress metabolic effects, inflammatory responses, and oxidative stress in diet-induced obese mice (DIO).

## 2. Materials and Methods

### 2.1. Diet and Characterization of Animals

Four-week-old C57BL/6J male mice were used for the experiments; they were maintained at 70% humidity, 20 ± 2 °C temperature, and exposed to a 12 h light–dark cycle. All experimental procedures were performed in accordance with the local ethics committee of the Extremo Sul Catarinense University (protocol number—007/2017-1). The mice were initially divided into 2 groups; one fed for 20 weeks with a standard diet for rodents (control, *n* = 20/group) (63% carbohydrates, 26% proteins, and 11% lipids—relative to calories, which corresponds to approximately 3.3 kcal/g—Puro Lab 22PB; Porto Alegre, RS, Brazil), and the other fed for 20 weeks with a high-fat diet (DIO, *n* = 20/group) (26% carbohydrates, 14.9% proteins, 59% lipids—relative to calories, which corresponds to approximately 5.35 kcal/g; PragSoluções Serviços e Comércio Ltd.a, Jaú, SP, Brazil) (Table 1). The daily caloric intake was 12kcal/day in the control group and 14,63kcal/day in DIO.

Animals were weighed weekly. After IR was effectively induced, the animals were randomly distributed into 4 groups (*n* = 10): control (control), control + avocado oil (control + AO), diet-induced obesity (DIO), and diet-induced obesity + avocado oil (DIO + AO). The stages of the study are shown in Figure 1.

### 2.2. Avocado Oil Supplementation

The animals supplemented received cold pressed avocado oil (100% purity; Pazze Indústria de Alimentos LTDA, Panambi, RS, Brazil) (4 mL/kg of total body weight), by oral gavage, once daily for 90 days [16]. Control animals received drinking water by oral gavage at the same volume [17]. During the 90 days of supplementation, all animals continued to receive their respective diets. The kcal/day was equal among all groups and was not affected by AO supplementation. The avocado oil dose of 1.0 mL/250 g by gavage was used in several studies and showed no genotoxic effect [16,18,19].

### 2.3. Characterization of Avocado Oil by Gas Chromatography

Avocado oil (301 mg) was transesterified by ISO method 5509:1978 (21) to obtain the methyl esters of fatty acids. In a tube, the sample and 5 mL of *n*-heptane were added. The tube was vortexed for 1 min, following which 1 mL of a 0.5 mol L^−1^ NaOH methanol solution was added and then vortexed again for 1 min. The upper organic phase was collected, anhydrous sodium sulfate was added and the solution was filtered.

The fatty acid methyl esters were determined on a Shimadzu gas chromatograph (GC 2010-Plus, Shimadzu, Kyoto, Japan) with a split/splitless injector, AOC-i-20 auto-injector, and flame ionization detection. A fused silica capillary column was used (CP-SIL 88 for FAME; 100 m × 0.25 mm × 0.2 μm, Agilent Technologies, Palo Alto, CA, USA). The chromatographic conditions were as follows: Injection volume of 1.0 μL in split mode 1:100, injector temperature at 250 °C, detector temperature at 260 °C, initial oven temperature set to 100 °C and held for 5 min, increased from 4 °C/min to 200 °C, and waited 30 min after reaching the final temperature. Hydrogen was used as the entrainment gas with a flow rate of 1.0 mL/min and a pressure of 140.3 kPa. Quantification of the fatty acid methyl esters was done by area normalization and the concentrations were expressed in g per 100 g sample (Table 2) [20].

### 2.4. Insulin Tolerance Test

The fasting insulin tolerance test was performed on days 135 and 225 of the experiment. The feed was withdrawn over an 8 h period and then reintroduced for 1 h and withdrawn again after this period. The test was performed after 4 h of fasting. The first blood collection (tip of the tail and a blood drop used were para assessed for the glucose in glucosimetrs) represents the zero time of the test. Humulin NPH insulin (1 U/kg of body weight) was then injected intraperitoneally, and blood samples were collected from the tail after 5, 10, 15, 20, 25, and 30 min to determine blood glucose levels. The constant rate of glucose decay (k_ITT_) was calculated using the formula 0.693/t_1/2_. Glucose t_1/2_ was calculated from the least squares analysis of serum glucose concentration during the linear decay phase [21]. 

### 2.5. Behavioral Tasks

The open field test was used to evaluate spontaneous locomotor activity. The mice (*n* = 10 per group) were randomly placed into individual square wooden boxes (40 × 60 × 50 cm) that were positioned on the floor of a soundproof and diffusely illuminated room for 5 min. The locomotor (number of crossings = crossed the square with all four legs) and exploratory activities (number of grooming= lifting of the animal supported by the hind legs) were measured.

In the same cage used for the open field test, a recognition memory task was performed. On the training day, the mice were familiarized with two identical objects. After 90 min, the novel place test trial (short-term memory), and after 24 h, the long-term memory tasks were performed. In these tasks, one of the objects was taken out, and a new object that differed in shape, color, and texture was put in its place. Between trials, the arena and all objects were thoroughly cleaned with 10% ethanol. Each exploration was defined as an act in which the mouse would approach the object with the nose (within 1 cm), sniff, and touch the object with the tip of its nose and/or with its paws. 

The anxiety-like behavior was analyzed using the plus maze task. The apparatus consisted of two open arms (18 × 6 cm), two opposite closed arms (18 × 6 cm), and was raised 60 cm from the floor in a low-light environment (12 lx). The following parameters were analyzed for 5 min: the number of entries and the time spent in the open arms (entry was considered when the four legs of the animal were inside the arm). 

### 2.6. Adipose Fat Pads

After the experimental period, the animals were fasted for 12 h before euthanasia without anesthesia using decapitation by guillotine, and the epididymal, mesenteric, perirenal, and retroperitoneal adipose tissues were extracted and weighed to calculate the fat pads, expressed as the percentage of total body weight (g of fat/g of total body weight × 100).

### 2.7. Measurement of Serum Triglycerides Levels

Blood samples were collected from the eyes of the animals, and the serum was separated by centrifugation and stored at −20 °C. Animals were fasted 12 h before euthanasia. Triglycerides levels were determined in serum using Trinder enzymatic colorimetric assay according to the manufacturer’s specifications (Life Biotechnology, Belo Horizonte, Brazil).

### 2.8. Measurement of Inflammatory Markers and Neurotrophic Factors Level

The samples were homogenized (1:10) in lysis buffer (NaCl, MgCl2, KCl, 1.5 M Tris, triton, glycerol, orthovanadate, aprotinin, pyrophosphate, and phenylmethylsulfonyl fluoride) and frozen at −80 °C until analysis. The levels of IL-1β and TNF-α cytokines in the epididymal adipose tissues, quadriceps, liver, and hippocampus, as well as levels of BDNF (Brain-Derived Neurotrophic Factor) and β-NGF (β-Nerve Growth Factor) were measured using the ELISA method (R&D Systems). The absorbance reading was performed at 450 nm.

### 2.9. Oxidative Status

The epididymal adipose tissue, quadriceps, liver, and hippocampus were homogenized in PBS buffer (pH 7.4; 5 mg tissue/100 µL PBS).

### 2.10. 2′-7′-Dichlorofluorescin Diacetate (DCFH-DA) Oxidation Assay

The sample was incubated with 80 mM DCF-DA at 37 °C for 15 min. DCFH formation was monitored with excitation and emission wavelengths of 488 and 525 nm, respectively [22]. 

### 2.11. Nitric Oxide (NO) Production Levels

The production of nitric oxide was evaluated spectrophotometrically by the stable nitrite metabolite (NO_2_). Samples were incubated with Griess reagent (1% sulfanilamide in 0.1 mol/L HCl and 0.1% *N*-(1-naphthyl) ethylenediamine, dihydrochloride) at room temperature for 10 min, followed by analysis at 540 nm in the spectrophotometer. Nitrite content was calculated from the standard curve of sodium nitrite (NaNO_2_)—0 to 200 nM. The results are expressed as μmol nitrite/mg protein [23].

### 2.12. Carbonyl Level in Proteins

The samples were precipitated with 20% trichloroacetic acid, and the proteins were dissolved in dinitrophenylhydrazine. Afterwards, the samples were redissolved in 6 M guanidine. The carbonyl content was determined spectrophotometrically at 370 nm using a 22,000 M absorbance coefficient [24].

### 2.13. Evaluation of Antioxidant Enzymes

Activity of the SOD enzyme (0.0024 mg/mL) was estimated by inhibiting the auto-oxidation of adrenaline and analyzed at a wavelength of 480 nm [25]. CAT enzyme activity was estimated by calculating the consumption rate of 10 mM hydrogen peroxide analyzed at a wavelength of 240 nm and expressed as U/mg of protein [26].

### 2.14. Protein Content Determination

Bovine serum albumin was used as a standard to assess the total protein content of the samples [27].

### 2.15. Statistical Analysis

The results are expressed as the mean ± standard deviation. The data were tested for normality (Shapiro–Wilk test) and equality of variance (Levene’s test) and analyzed statistically using a one-way ANOVA test. This was followed by post hoc analysis using Tukey’s test to compare all pairs of columns in GraphPad Prism 7.0. Statistical differences between groups were considered significant at *p* < 0.05.

## 3. Results

### 3.1. Effects of DIO and Avocado Oil Supplementation on Glucose Levels, Insulin Sensitivity, Body Weight/Adiposity, and Serum Triglyceride Levels

The insulin tolerance test pre-supplementation showed that the DIO group had lower insulin sensitivity than the control group (Figure 2A), showing installation of insulin resistance. The pos-supplementation showed the glucose levels after 4 h of fasting were higher in the DIO and DIO + AO groups than in the control and control + AO groups (Figure 2B). In the k_ITT_ analysis, a lower decay constant was observed in the DIO group compared to that in the control and DIO + AO groups (Figure 2C), and the AUCs in the DIO and DIO + AO groups were higher than those in the control groups (Figure 2D). 

DIO and DIO + AO induced a higher weight gain throughout the experiments; however, just after the start of AO supplementation, the DIO + AO group showed a decrease in body weight; in addition, in the last 4 weeks, both groups had similar body weights (Figure 2E). The percentage of the epididymal, mesenteric, retroperitoneal, and perirenal fat pads was higher in the DIO and DIO + AO groups than in the control and control + DIO groups (Figure 2F–I). The DIO group had higher serum triglyceride levels than the other groups (Figure 2J). 

### 3.2. Effects of DIO and Avocado Oil Supplementation on Locomotion, Cognition, and Anxiety

Recognition memory improved in animals in the control group, but only in the long-term memory test. Animals in the control + AO and DIO + AO groups showed improved performance in the short- and long-term memory tests. Animals in the DIO group did not show any difference concerning the recognition index (Figure 3A). The animals in the DIO group showed a higher number of crossings than those in the control group (Figure 3B). There were no differences between the groups in the number of crossings (Figure 3C). There were no differences between groups concerning time or number of entries in the open arms in the plus maze test (Figure 3D,E). There was no change in the levels of BDNF (Figure 3F) and NGF-β (Figure 3G) in the hippocampus of all the groups. 

### 3.3. Effects of DIO and Avocado Oil Supplementation on the Epididymal Adipose Tissue

In the epididymal adipose tissue, the DCFH level was higher in the DIO group than in the DIO + AO group (Figure 4A). The nitrite concentration was higher in the DIO group than in the control and DIO + AO groups (Figure 4B). The carbonyl content of proteins was higher in the DIO group than that in the control and control + AO groups (Figure 4C). The antioxidant activities of SOD and CAT were increased in the DIO + AO group (Figure 4D) and control + AO and DIO + AO groups (Figure 4E), respectively, compared to those in the other groups. The levels of both pro-inflammatory cytokines, IL-1β and TNF-α, were increased in the DIO group compared to those in the control and control + AO groups (Figure 4F,G, respectively).

### 3.4. Effects of DIO and Avocado Oil Supplementation in the Quadriceps

In the quadriceps, the DCFH levels were higher in the DIO group than in the other groups (Figure 5A), and the nitrite concentrations were higher in the DIO group than in the control and control + AO groups (Figure 5B). There was no change in the carbonyl content of proteins in any of the groups (Figure 5C). SOD activity was higher in the DIO + AO group than in the other groups (Figure 5D). There was no change in the CAT activity in any of the groups (Figure 5E). The levels of the pro-inflammatory cytokines IL-1β and TNF-α were higher in the DIO group than in the other groups (Figure 5F,G, respectively).

### 3.5. Effects of DIO and Avocado Oil Supplementation on the Liver

In the liver, the DCFH level was higher in the DIO group than in the DIO + AO group (Figure 6A). The nitrite concentration was higher in the DIO group than in the control, control + DIO, and DIO + AO groups (Figure 6B). The carbonyl content of proteins was higher in the DIO group than in the other groups (Figure 6C). The activity of the antioxidant enzyme SOD was increased in the control + AO group compared to that in the other groups (Figure 6D). CAT activity was increased in the control + AO and DIO + AO groups compared to that in the control and control + AO groups (Figure 6E). The levels of the pro-inflammatory cytokines IL-1β and TNF-α were not different among the groups (Figure 6F,G, respectively).

### 3.6. Effects of DIO and Avocado Oil Supplementation on the Hippocampus

In the hippocampus, there was no change in the DCFH levels in any group (Figure 7A). The nitrite concentration was higher in the DIO and DIO + AO groups compared to that in the control and control + DIO groups (Figure 7B). The carbonyl content of proteins was decreased in the control + AO and DIO + AO groups compared to that in the other groups (Figure 7C). The activity of the antioxidant enzyme SOD was decreased in the control + AO and DIO groups compared to the other groups (Figure 7D). There was no change in the CAT activity in any of the groups (Figure 7E). The levels of the pro-inflammatory cytokine IL-1β were increased in the DIO group compared to the other groups (Figure 7F). The levels of the pro-inflammatory cytokine TNF-α were increased in the DIO and DIO + AO groups compared to the control groups (Figure 7G).

## 4. Discussion

Poor diet and a sedentary lifestyle contribute to the recent increase in obesity rates. Herbal compounds and some foods seem to be effective treatments in the fight against obesity and its complications [1]. In this study, AO treatment-induced changes in the blood, cognition, liver, muscle, adipose, and hippocampus of previously obese animals. Pharmacological treatments for obesity are not completely effective and have negative side effects [28], and AO could be a promising strategy.

AO supplementation (5%, 10%, 20%, or 30% for 8 weeks) improved glucose sensitivity in animals fed a sucrose-rich diet [29]. In our study, the DIO group presented with IR, and supplementation with AO improved insulin sensitivity. The effects of AO on insulin sensitivity could be due to its high oleic acid concentration (almost 50%), which facilitates glycemic control in patients with diabetes mellitus [30]. Oleic acid favors the infiltration of m2-type macrophages into adipose tissue, which has an anti-inflammatory profile [31]. DIO increases body weight and adiposity in C57BL/6J mice [32], which is consistent with our results. AO supplementation did not modify these parameters at the end of the experiment. However, after the start of AO supplementation, the DIO + AO group decreased their body weight, which was not explained by the changes in the daily kcal intake, which were equal in all groups. Supplementation with avocado hydroalcoholic extract (100 mg/kg body weight) reduced body weight gain and adiposity index in DIO rats [33]. Usually, avocado oil or hydroalcoholic extract supplementation starts concomitantly with the beginning of the diet protocol in rodents, whereas in our study, AO treatment started after 20 weeks of a high-fat diet (when the animals were already obese). 

High triglyceride levels are associated with heart diseases [34]. DIO increased serum triglyceride levels, and AO supplementation reversed this effect. Non-alcoholic hepatic steatosis is associated with chronic systemic inflammation and oxidative stress [1,35] and AO supplementation decreased the presence of cell infiltrates with adipocyte morphology in the liver. A higher intake of monounsaturated fatty acids (MUFAs), as observed in the AO, is associated with lower triglyceride levels and better glycemic control [36], positively impacting fat distribution, thereby increasing fat deposition in the adipose tissue rather than in the liver [37].

Adipose tissue in obese subjects has high levels of inflammatory molecules [38]. However, subclinical inflammation in obesity can be present in other tissues, such as the liver [39]. DIO mice expressed higher TNF-α and IL-1β levels in the adipose tissue after 24 weeks and in the liver after 40 weeks of consuming DIO, indicating that IR was established before the development of liver inflammation [40]. We found that the DIO group showed increased TNF-α and IL-1β levels in the adipose tissue and quadriceps but not in the liver, which potentially occurred because of the duration of the DIO protocol. Obesity-induced increase in IL-1β levels, mostly in diabetes patients, and the improvement in insulin sensitivity was associated with decrease in cytokine levels [41]. We showed that AO had an anti-inflammatory effect on the adipose tissue and muscle and increased insulin sensitivity, showing a possible mechanism for AO effects even no changes in body weight.

In addition to its association with IR, inflammation is linked to oxidative stress [39]. Nutraceuticals, such as ginger and avocado, can attenuate ROS and RNS [42]. In our study, after AO supplementation for 90 days, the metabolic biomarkers of oxygen and nitrogen in the adipose tissue were reduced in DIO animals and improved in control animals. The decline in ROS generation was observed in the kidneys of diabetic rats supplemented with 4 mL/kg of AO for 3 months [18]. Besides causing an increase in reactive species in the adipose and skeletal muscle tissues, DIO in rodents also culminates in increased ROS production and enzyme iNOS expression in the liver [43]. In our study, although DIO increased the DCFH levels in the quadriceps, it did not increase its levels in the liver; furthermore, there was a significant increase in nitrite production in both tissues, which may indicate improved RNS formation. In contrast, supplementation with AO in the DIO group reduced the nitrite concentration and led to lower oxidation of DCFH in the liver and quadriceps. Excess nitric oxide could induce a higher formation of peroxynitrite, which causes protein damage through carbonylation [44]. 

Co-incubation of palmitic acid-exposed beta cells activates antioxidant defense pathways involved in protein folding, protecting cells against oxidative stress [45]. The epididymal adipose and liver tissues of animals in the DIO group showed higher protein damage, which was reversed after supplementation with AO. The aqueous extract of avocado is reported to possess hepatoprotective effects in a liver toxicity model [14].

Obesity decreases SOD and GPX antioxidant enzymes activity in the epididymal adipose tissue, thus inducing protein carbonylation [46]. We found no differences in the activity and levels of antioxidant enzymes in DIO animals. In contrast, obese animals that received AO showed increased SOD and CAT activities in a tissue-specific manner, demonstrating the antioxidant potential of AO, in the adipose tissue, quadriceps, and liver. It is suggested that the antioxidant capacity of AO is attributed to its α-tocopherol and β-sitosterol content, which improve SOD and CAT antioxidant enzyme activities [47]; both these compounds were prevalent in the AO oil used in this study. AO attenuates the deleterious effects of oxidative stress in the liver of rats by improving mitochondrial function [19]. Additionally, AO is rich in carotenoids, which can remove cellular O_2_^–^, suggesting that the carotenoids in AO could also play a protective role against oxidative damage [48]. The modulation of anti-oxidant enzymes, even in control animals, could address that OA could be beneficial for preventing other metabolic or chronic diseases by reducing the generation of oxidative stress.

DIO and Western diets not only impair physiological conditions, but also alter brain regions that control cognition [4] and can induce anxiety and depressive behavior [49]. Recognition memory and spatial learning were both decreased in DIO mice [50], while no signs of anxiety were observed [51]. DIO impaired recognition memory and AO supplementation reversed this effect in obese animals. However, the mood parameters were not affected by the treatments. Daily consumption of avocado, for 6 months, improves memory, spatial working memory, sustained attention, and efficiency in approaching a problem in healthy adults [52]. AO supplementation for 90 days in diabetic animals decreases ROS production and damages proteins by improving the respiratory chain complex III, in addition to increasing brain antioxidant activity [16]. In the hippocampus, IL-1β was increased by DIO, and AO was able to revert this; however, AO did not affect TNF-α levels in DIO animals, thus indicating the limitations of the anti-inflammatory effects of AO in the brain. Although no evaluations were performed to determine AO effects in obese animals, olive oil supplementation, which has physicochemical characteristics similar to AO, reverses hypothalamic inflammation and improves the anti-inflammatory action exerted in the hippocampus due to the abundant presence of MUFA in its composition [52]. 

## 5. Conclusions

In summary, our results show that DIO induced changes in all the tissues evaluated. AO supplementation promotes benefits at the molecular level in previously obese mice, even without causing a change in body weight and adiposity. Although a limitation of our study was that we were not able to propose a specific mechanism underlying AO protection in obesity, we believe that AO modulates intracellular distinct pathways in many tissues. AO supplementation decreased serum triglyceride levels. In terms of recognition memory, AO improved short- and long-term memory in all groups. Moreover, it reduced oxidative damage to the proteins in all tissues. Furthermore, it improved antioxidant enzyme activity and decreased the inflammatory status in the epididymal adipose and skeletal muscle tissues.

## Figures and Tables

**Figure 1 nutrients-14-02906-f001:**
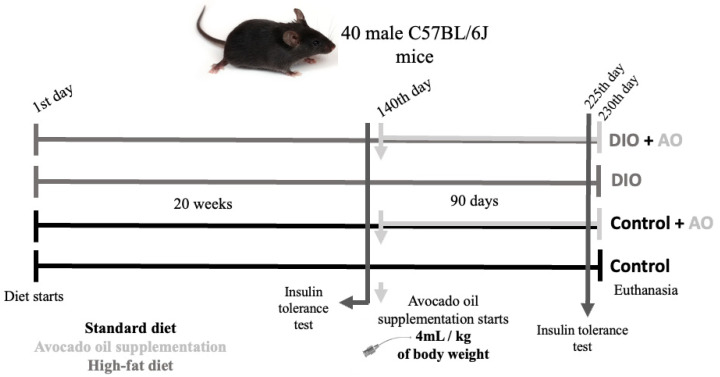
*Scheme depicting the experimental flowchart.* Animals were placed in control or diet-induced obesity (DIO) groups for 20 weeks and an insulin tolerance test was performed. After both groups received avocado oil (AO) supplementation by gavage for 90 days, another insulin tolerance test was performed, animals were euthanized, and tissue samples were collected.

**Figure 2 nutrients-14-02906-f002:**
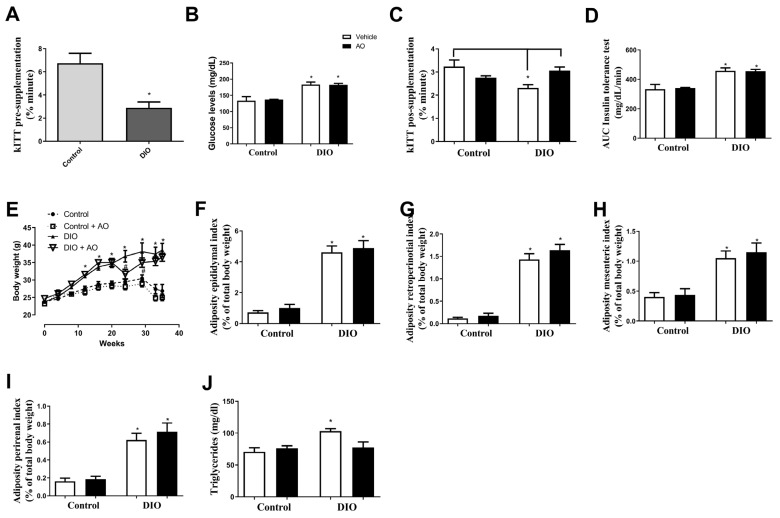
*Effects of avocado oil (AO) supplementation on i**nsulin sensitivity, fat pad index, and serum triglyceride and lipoprotein levels**in mice with diet-induced obesity (**DIO).* (**A**) Rate of glucose decay (k_ITT_) pre-supplementation after insulin intraperitoneal injection (* DIO < control, *p* < 0.05). (**B**) Fasting blood glucose levels (* DIO and DIO + AO > control and control + AO, *p* < 0.05). (**C**) Rate of glucose decay (k_ITT_) after insulin intraperitoneal injection (* DIO < control and DIO + AO, *p* < 0.05). (**D**) Area under the curve (AUC) of the insulin tolerance tests (* DIO and DIO + AO > control and control + AO, *p* < 0.05) (**E**) Body weight (* DIO and DIO + AO > control and control + AO; ^#^ DIO + AO < DIO, *p* < 0.05). (**F**) Epididymal fat pad (* DIO and DIO + AO > control and control + AO, *p* < 0.05). (**G**) Retroperitoneal fat pad (* DIO and DIO + AO > control and control + AO, *p* < 0.05). (**H**) Mesenteric fat pad (* DIO and DIO + AO > control and control + AO, *p* < 0.05). (**I**) Perirenal fat pad (* DIO and DIO + AO > control and control + AO, *p* < 0.05). (**J**) Serum triglyceride levels (* DIO > control, control + AO and DIO + AO, *p* < 0.05). The results are expressed as the mean ± standard deviation. The data were analyzed statistically using a one-way ANOVA test followed by post hoc analysis using Tukey’s test to compare all pairs of columns. Data were collected from 5 to 7 animals per group. * > control and control + AO.

**Figure 3 nutrients-14-02906-f003:**
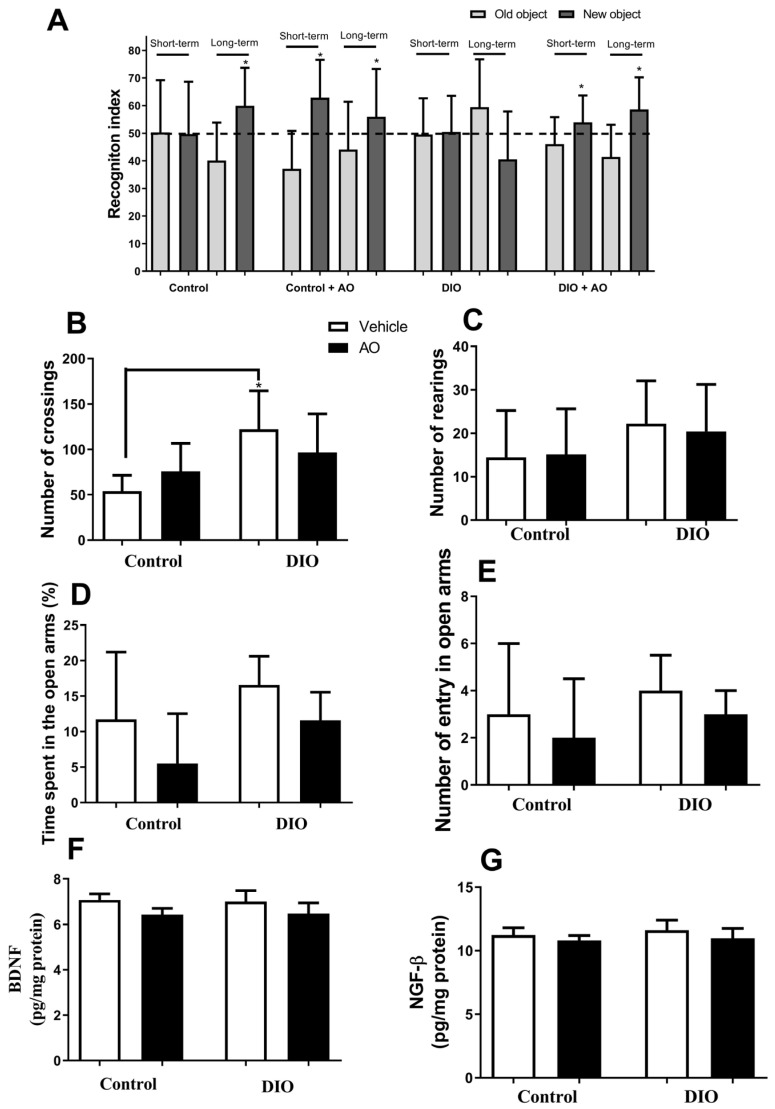
*Effects of avocado oil (AO) supplementation on cognitive performance and mood parameters.* (**A**) Recognition memory (* index recognition, new object > old object). (**B**) Number of crossing (* DIO > control, *p* < 0.05). (**C**) Number of rearings. (**D**) Time spent in the open arms. (**E**) Number of entries in open arms. (**F**) BDNF level in the hippocampus. (**G**) NGF-β level in the hippocampus. The results are expressed as the mean ± standard deviation. The data were analyzed statistically using a one-way ANOVA test followed by post hoc analysis using Tukey’s test to compare all pairs of columns. Data were collected from 5 to 7 animals per group. Data for behavioral tasks were collected from 10 animals per group.

**Figure 4 nutrients-14-02906-f004:**
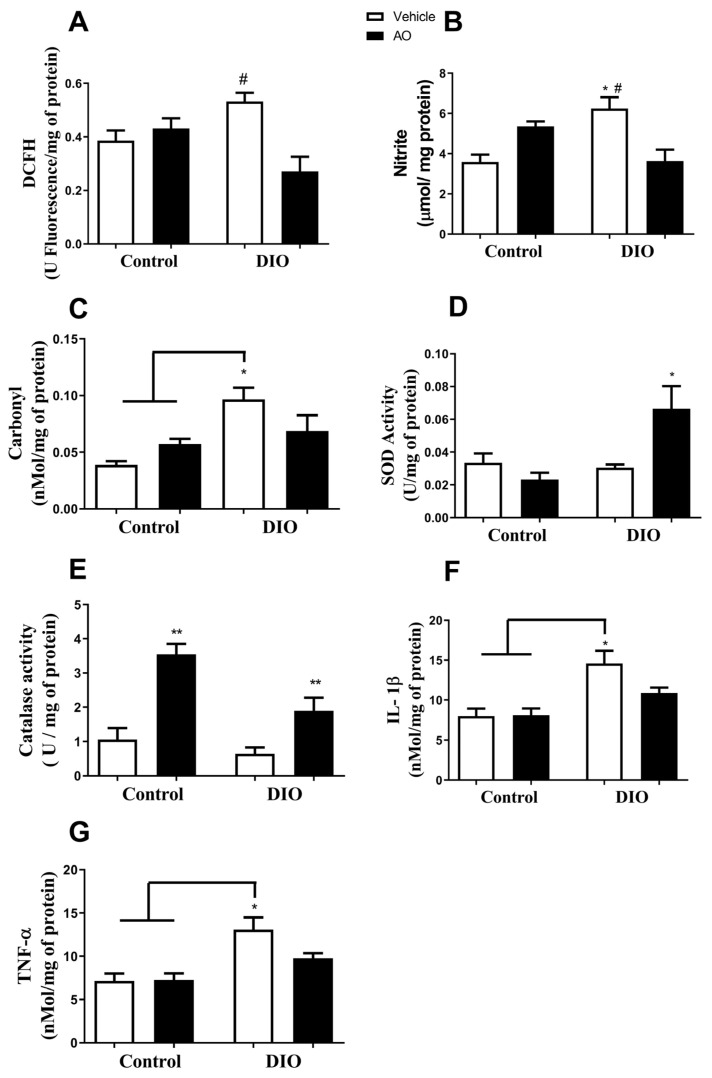
*Effects of avocado oil (AO) supplementation on reactive species production, oxidative damage, antioxidant enzyme activity, and cytokine levels in the epididymal adipose tissue of mice with diet-induced obesity (DIO).* (**A**) 2′-7′-Dichlorofluorescin (DCFH) levels (^#^ DIO > DIO + AO, *p* < 0.05). (**B**) Nitrite levels (* DIO > control; ^#^ DIO > DIO + AO, *p* < 0.05). (**C**) Carbonyl content of proteins (* DIO > control and control + AO, *p* < 0.05). (**D**) Superoxide dismutase (SOD) activity (* DIO + AO > control, control + AO and DIO, *p* < 0.05). (**E**) Catalase (CAT) activity (** control + AO and DIO + AO > control and DIO, *p* < 0.05). (**F**) Interleukin 1β (IL-1β) levels (* DIO > control and control + AO, *p* < 0.05). (**G**) Tumor necrosis factor-α (TNF-α) levels (* DIO > control and control + AO, *p* < 0.05). The results are expressed as the mean ± standard deviation. The data were analyzed statistically using a one-way ANOVA test followed by post hoc analysis using Tukey’s test to compare all pairs of columns. Data were collected from 5 to 7 animals per group. * > control. ^#^ > DIO + AO.

**Figure 5 nutrients-14-02906-f005:**
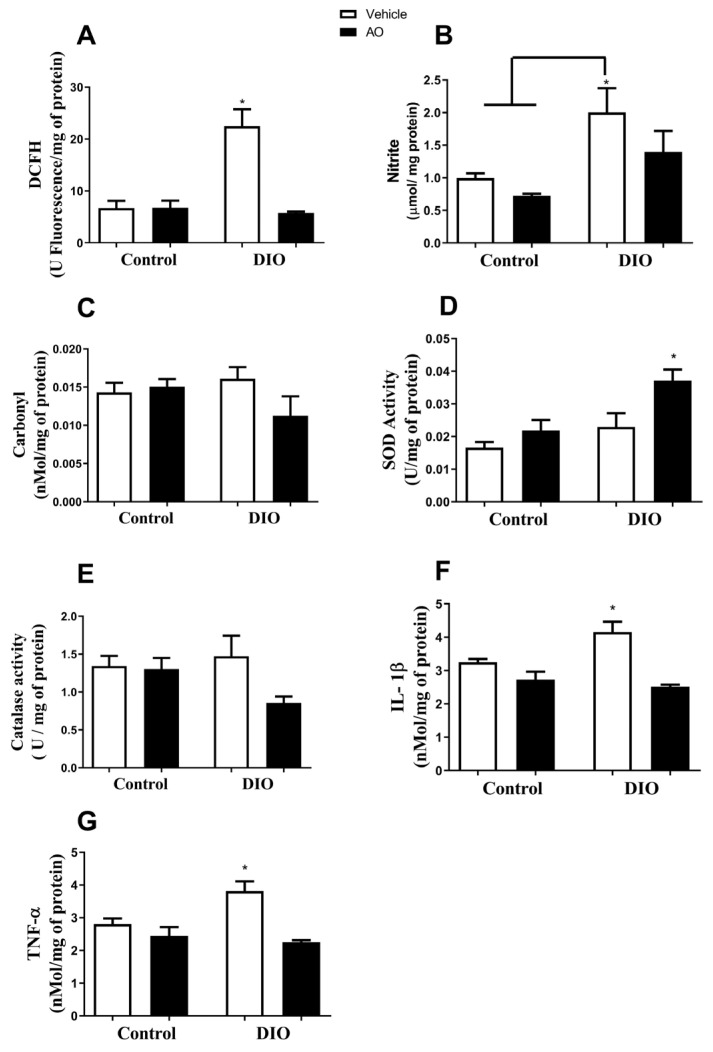
*Effects of avocado (AO) supplementation on reactive species production, oxidative damage, antioxidant enzymes activity and cytokine levels in the quadriceps tissues of mice with diet-induced obesity (DIO*). (**A**) 2′-7′-Dichlorofluorescin (DCFH) levels (* DIO > control and control + AO, *p* < 0.05). (**B**) Nitrite levels (* DIO > control and control + AO, *p* < 0.05). (**C**) Carbonyl content. (**D**) Superoxide dismutase (SOD) activity (* DIO + AO < control, control + AO and DIO, *p* < 0.05). (**E**) Catalase (CAT) activity. (**F**) Interleukin 1β (IL-1β) levels (* DIO > control, control + AO and DIO + AO, *p* < 0.05). (**G**) Tumor necrosis factor-α (TNF-α) levels (* DIO > control, control + AO and DIO + AO, *p* < 0.05). The results are expressed as the mean ± standard deviation. The data were analyzed statistically using a one-way ANOVA test followed by post hoc analysis using Tukey’s test to compare all pairs of columns. Data were collected from 5 to 7 animals per group. * > control and control + AO.

**Figure 6 nutrients-14-02906-f006:**
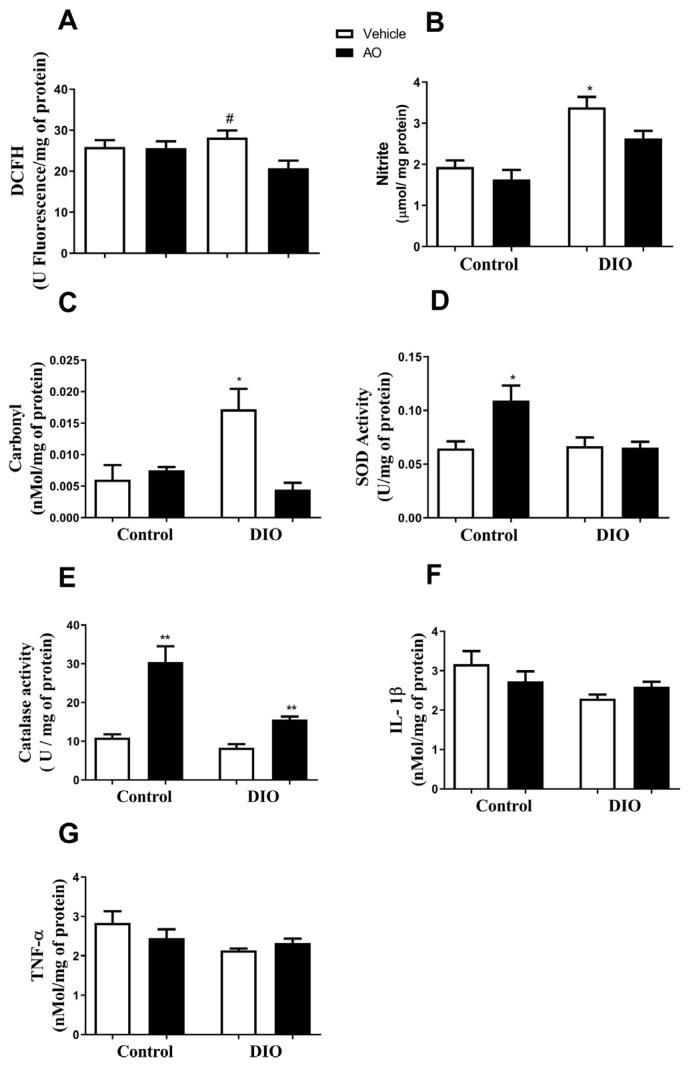
*Effects of avocado oil (AO) supplementation on reactive species production, oxidative damage, antioxidant enzymes activity and cytokines levels in the liver of mice with diet-induced obesity (DIO).* (**A**) 2′-7′-Dichlorofluorescin (DCFH) levels (^#^ DIO > DIO + AO, *p* < 0.05). (**B**) Nitrite levels (* DIO > control, control + AO and DIO + AO, *p* < 0.05). (**C**) Carbonyl content of proteins (* DIO > control, control + AO and DIO + AO, *p* < 0.05). (**D**) Superoxide dismutase (SOD) activity (* control + AO > control, DIO and DIO + AO, *p* < 0.05). (**E**) Catalase (CAT) activity (** control + AO and DIO + AO > control and DIO, *p* < 0.05). (**F**) Interleukin 1β (IL-1β) levels. (**G**) Tumor necrosis factor-α (TNF-α) levels. The results are expressed as the mean ± standard deviation. The data were analyzed statistically using a one-way ANOVA test followed by post hoc analysis using Tukey’s test to compare all pairs of columns. Data were collected from 5 to 7 animals per group. * > control, control + AO and DIO + AO. ** > control and DIO. ^#^ > DIO + AO.

**Figure 7 nutrients-14-02906-f007:**
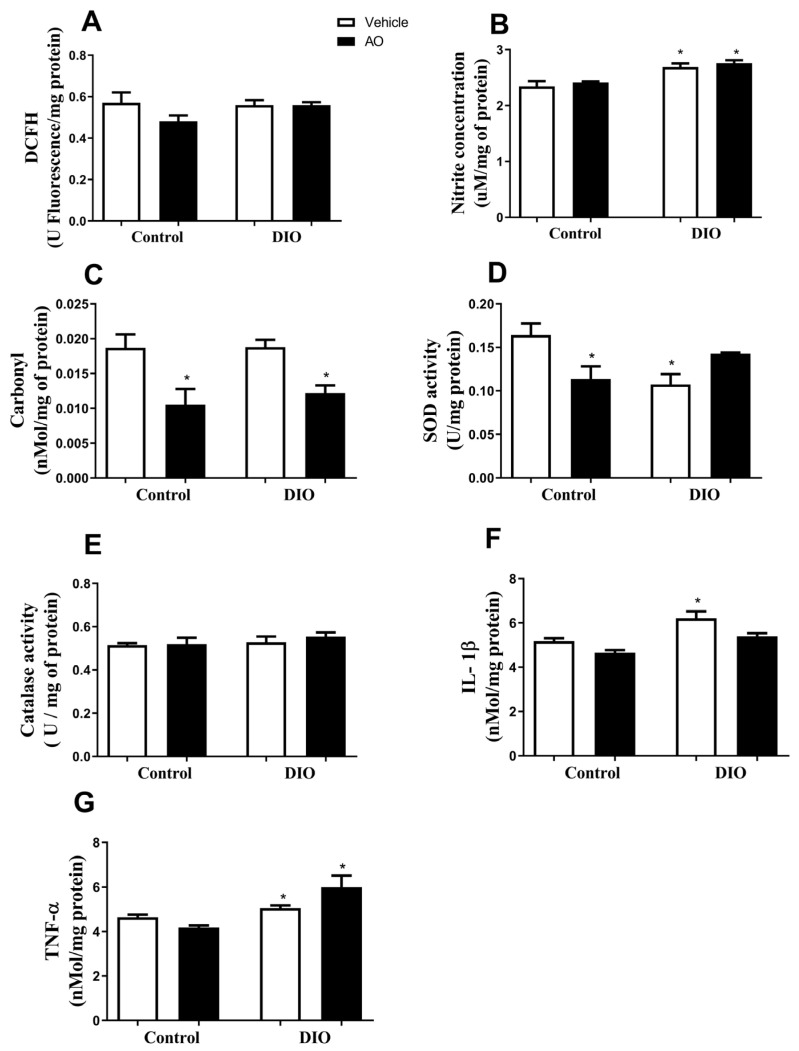
*Effects of avocado oil (AO) supplementation on reactive species production, oxidative damage, antioxidant enzymes activity and cytokines levels in the hippocampus of mice with diet-induced obesity (DIO).* (**A**) 2′-7′-Dichlorofluorescin (DCFH) levels. (**B**) Nitrite levels (* DIO > control, control + AO and DIO + AO, *p* < 0.05). (**C**) Carbonyl content of proteins (* DIO > control, control + AO and DIO + AO, *p* < 0.05). (**D**) Superoxide dismutase (SOD) activity (* control + AO > control, DIO and DIO + AO, *p* < 0.05). (**E**) Catalase (CAT) activity. (**F**) Interleukin 1β (IL-1β) levels. (**G**) Tumor necrosis factor-α (TNF-α) levels. The results are expressed as the mean ± standard deviation. The data were analyzed statistically using a one-way ANOVA test followed by post hoc analysis using Tukey’s test to compare all pairs of columns. Data were collected from 5 to 7 animals per group. * > control, control + AO and DIO + AO.

**Table 1 nutrients-14-02906-t001:** Diet composition.

	High-Fat Diet			Standard Diet	
	g/1000 g	kcal/1000 g		g/1000 g	kcal/1000 g
Corn starch	147.5	590	Carbohydrates	530	2120
Casein	200	800	Protein	220	880
Dextrinized starch	100	400	Lipids	40	360
Sucrose	100	400	Phosphorus	8	-
Soy oil	40	360	Mineral mix	80	-
Microcrystallized cellulose	50	-	Calcium	10	-
Mineral mix AIN93G	35	-	Fiber	70	-
Vitamin mix AIN93	10	-			
L-cystine	3	-			
Choline bitartrate	2.5	-			
Lard	312	2808			

**Table 2 nutrients-14-02906-t002:** Bioactive compounds present in Pazze avocado oil.

Identification	Concentration of Fatty Acids (%/)
C16:0 Methyl palmitate	7.68
C18:0 Methyl stearate	2.79
C16:1 (*cis*-9) Methyl palmitoleate	2.00
C18:1 (*cis*-9) Methyl oleate (omega 9)	48.73
C18:2 (*cis*-9,12) Methyl linoleate (omega 6)	37.10
C18:3 (*cis*-9,12,15) Methyl linolenate (omega 3)	3.70
Identification	Concentration of most important phytosterols (%)
α-Tocopherol	75.00
β-Sitosterol	6.00

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
