# Peer review of "Effects of Avocado Oil Supplementation on Insulin Sensitivity, Cognition, and Inflammatory and Oxidative Stress Markers in Different Tissues of Diet-Induced Obese Mice"

_nutrients, 2022, doi:10.3390/nu14142906_

Round 1

Reviewer 1 Report

In their “Answers to reviewers” the authors were not able to rebut exhaustively my observations (in some cases, they just limited to remove unclear results without trying to explain them, e.g. blood lipid trend) and no significant changes have been made on the manuscript that therefore also in the revised version is scientifically inconsistent. Most of all, as for the previous version, the main problem is that data are not strong and congruent enough to support the authors’ assumption, namely the efficacy of avocado oil for the treatment of obesity and its outcomes.

Please also note that English is still poor and that also some minor flaws have been not corrected properly (e.g. resveratrol is still classified as a phytosterol).

Author Response

Dear reviewer

Reviewer 2 Report

In this study, Schérolin de Oliveira Marques et al. evaluated the effect of AO supplementation on obese mice in the adipose tissue, muscle, liver, and hippocampus.  From my point of view, I consider this study interesting and methodologically well executed. However, there are some aspects that deserve clarification.

-One aspect in method section is not very clear. If two insulin tolerance tests were performed, why did you only show the results of the second test? was it not preferable for the KITT parameter to perform a repeated measured ANOVA between the basal point and the end of follow up? Did you not run it because the sample was small? in this case a Kruskal Wallis test can be used between the Δ of the two points.

-The figure 2D is difficult to read. I recommend showing It separately. Furthermore I believe there is a discrepancy between the text and the figure 2D. In the text is reported “just after 214 the start of AO supplementation, the DIO + AO group showed a decrease in body weight 215 for 8 weeks; in the last 4 weeks, both groups had similar body weights”. But in the legend to the figure 2D it is written that there is a statistically significant difference between DIO and DIO-AO. What does this mean? Did you use a repeated measures ANOVA using 0-10-20-30-40 as time point and did you find a statistically significant difference? Then you performed a Kruskal Wallis test on day 90 and the final weight differences weren't statistically significant? this part should be clarified.

-In a rather recent original paper it was shown that in obese people serum IL-1β and Caspase-1 levels were inversely correlated with IR, and that Caspase-1 and IL-1β levels normalized after weight loss in subjects without diabetes (PMID: 32020775). In this murine model, do you believe that the contribution of OA on the reduction of IL-1b in adipose and muscle tissue is mainly due to weight reduction or the anti-inflammatory effect? it could be useful to describe in a few lines the parallelism between the animal model and the human model.

-There are some grammatical issues and typos that should be addressed. Some examples: page 2 row 69 the dot is missing; page 4 row 128 “glicose” etc.

Author Response

Dear reviewer

best regards

Reviewer 3 Report

This study provides interesting information about effects of avocado oil to reverse the heightened insulin resistance, markers of inflammation, and behavioral changes in an animal model of diet induced obesity. 

The data are solid, but the caloric intake (kcal/day) data mentioned in line 102 should be shown.

The paper would be improved by a paragraph to address potential mechanisms.

The effects on catalase and SOD are interesting.  Please add a paragraph discussing what is known regarding regulation of catalase and SOD expression and how avocado oil might be affecting this.

It is important to address whether/if avocado oil has different effects from those previously reported for oleic acid,  and olive oil.  It would also be helpful to know whether hydroalcoholic extract of avocado (as cited in the paper) contains oleic acid or has a separate group of molecules that could mediate an effect separate from oleic acid.

Very minor:

Please check the English for line 128.

Line 199:  which protein assay was used?

Figure 2C:  seems like this should not be arbitrary units

Line 335:  I don’t think adjuvant is the correct term (same for line 426)

Author Response

Dear reviewer

besta regards

Round 2

Reviewer 1 Report

No significant improvement has been made, thus I confirm my previous judgment and reject the manuscript. Please note that are still too many English mistakes.

Author Response

Dear reviewer

I respect so much your opinion. True, that is a fair point, but I have to say I disagree. I think a promissor study. 

best regards

This manuscript is a resubmission of an earlier submission. The following is a list of the peer review reports and author responses from that submission.

Round 1

Reviewer 1 Report

Remarks to the article

 Effects of avocado oil supplementation on insulin sensitivity, 2 cognition, and inflammatory and oxidative stress markers in 3 different tissues of DIO mice

The manuscript is pretty well designed but following questions should be answered.

In Material and methods section information concerning the used animal model should be add. What is the DIO mice how this animal develops obesity.

Authors have used lard in diet, why the butter was not used? Butter contains more cholesterol about 250 mg/100 g and more saturated fatty acids. Lard about 100 mg cholesterol/100 g and about 45% of monounsaturated fatty acids. Please explain your choice.  this.

Please add information how blood was taken  for insulin tolerance test in subchapter “Insulin tolerance test” .

Table 2 contains only information about avocado oil. Why the lard composition is not presented in this table?

Results

Figure 2  and 4-7 should be changed for tables.

Discussion

Discussion should be more focused on the DIO deficient mice . The mechanism of development of obesity should be taken into consideration.

Reviewer 2 Report

The present manuscript would account for the potential properties and applicability of avocado oil (AO) against obesity and its adverse effects. In sustaining that, obese mice submitted or not to AO supplementation have been employed, but almost data obtained in this model are inconsistent and/or not significant and are thus not able to support the authors’ assumption. 

In particular, in none of the tissues examined there is a common trend in the anti-oxidative and anti-inflammatory activities of AO which can actually differentiate animals that received it or not, whether they were obese or not.  On the contrary, its supplementation increased some anti-oxidant parameters in controls (e.g. NO, DCFH, carbonyl in epididymial adipose tissue, but not in liver, quadriceps, and hippocampus, superoxide dismutase in quadriceps and liver but thee enzyme decreased in epididimyal tissue, where catalase increased instead!), but decreased inflammatory cytokines; as regards obese mice, AO decreased NO, DCFH, and carbonyl in many cases, but modulation of anti-oxidant enzymes is less homogeneous as well as of cytokines.

Please also note that also some outcomes did not significantly differentiate between control and obese mice, regardless of AO supplementation (e.g. cognitive function, LDL cholesterol). Of note, it seems strange that HDL cholesterol is higher in animals fed the high-fat diet.

Based on all this evidence, how can one say that AO is a promising agent for obesity treatment?

Maybe, as the authors write, the model adopted has some limits since AO supplementation began after 22 weeks, when obesity has already occurred. But, in this case, why did they chose this protocol? Why did they not consider for their comparison also a group for which the high-fat diet and AO supplementation started at the same time? Moreover, why did they not analyze anti-oxidant and anti-inflammatory parameters also in blood? Maybe, they could be more representative for the AO activity. Also, analysis of some molecular signaling could have given some suggestions at this regard.

As minor observations:

  • “Materials and methods” are quite superficial and need to be more exhaustive, in particular methods for BDNF (what is it?) and NGFbeta are missing;
  • in fig. 2 is not clear at which times parameters have been measured;
  • results in fig. 3 are not clear for people not skilled in this field, the authors should better explain what graphics represent;
  • references for lines 335-337 and lines 356-358 are missing (in fact, in the latest, ref. 33 does not seem correct);
  • quotations 7, 9, 34, 53 do not seem proper;
  • tocopherol is not a phytosterol!!!

Finally, there are also English mistakes and/or some sentences are meaningless (e.g. lines 52-54, 116-117, 132-142).

In conclusion, the work in the submitted form is not able to give exhaustive and in-depth information.